# FedSKU: Defending Backdoors in Federated Learning Through Selective Knowledge Unlearning

## Abstract

Federated Learning (FL) has been found to be vulnerable to backdoor attacks, which involve an adversary uploading manipulated model parameters to deceive the aggregation process. Although several defenses have been proposed for backdoor attacks in FL, they are typically coarse-grained, as all of the methods process the uploaded model as a whole by either removing them or adding noises. In this paper, we propose a more fine-grained approach by further decomposing the uploaded model into malicious triggers and useful knowledge, which can be separately processed for improved performance. Specifically, our approach, called FedSKU, enables backdoor defense through **S**elective **K**nowledge **U**nlearning. We draw inspiration from machine unlearning to unlearn the malicious triggers while preserving the useful knowledge to be aggregated. Consequently, we accurately remove the backdoor trigger without sacrificing any other benign knowledge embedded in the model parameters. This knowledge can be further utilized to boost the performance of the subsequent aggregation. Extensive experiments demonstrate its superiority over existing defense methods [1].

## 1 Introduction

Federated Learning (FL), which enables collaborative learning while preserving privacy, has garnered considerable attention from both academia and industry (McMahan et al., 2017; Niu et al., 2020). Although being widely adopted, FL is vulnerable to backdoor attacks, where an adversary uploads manipulated model parameters to deceive the FL process (Bagdasaryan et al., 2020; Bhagoji et al., 2019; Sun et al., 2019). Different from traditional backdoor attacks that target at misleading a single model, in FL, the primary goal is to introduce some malicious behaviors in one or several clients, which can subsequently affect all participants. In other words, backdoor attacks in FL can have a broader negative impact, especially considering that there are usually a large number of clients involved in a system.

The research community has noticed the issue and several solutions have been proposed to defend FL backdoors. Typically there exists two types of defense ideas: The first one is *removal-based defense* (Blanchard et al., 2017), where a detection method is developed to accurately identify models originating from malicious clients, which are then directly removed. Another one is *noise-based defense* (Nguyen et al., 2022), which introduces some specially designed noises to mitigate the influence of backdoor. Despite being effective, we observe that existing defense methods are coarse-grained, as all of them process the uploaded model as a whole to remove or obfuscate. Our motivation is that, even when a client is detected as a backdoored one, its contributed model often retains valuable information, especially in the high non-IID settings where the local data in each client is unique. In addition, when the proportion of backdoored models gets higher, the loss of knowledge becomes increasingly severe if we adopt existing coarse-grained methods. Unlocking the potential knowledge of backdoored models is highly desirable in scenarios where accuracy performance is stringent or users want to achieve a trade-off between accuracy and attack success rate.

In this paper, we attempt to develop a more fine-grained approach to address the limitations of existing defense methods. Instead of discarding this entirely, we propose to isolate and utilize the benign

---

[1]Source code will be public after being accepted.

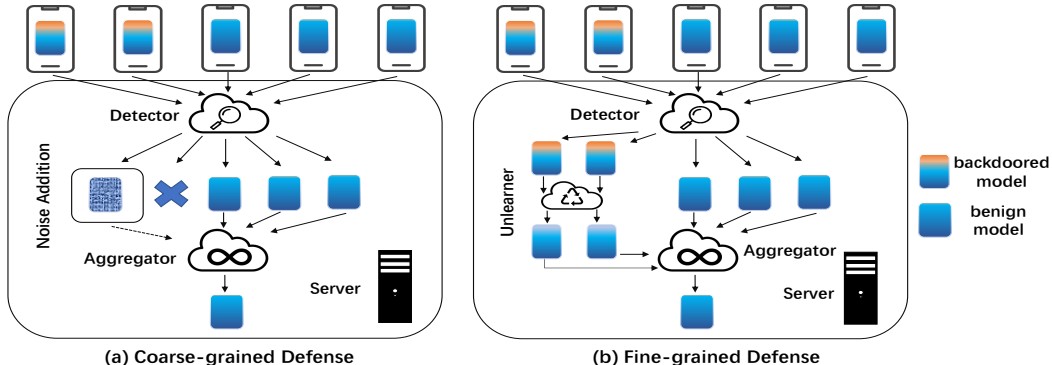

Figure 1: Comparison of the coarse-grained defense and our fine-grained defense.

knowledge inside the backdoored models. The key idea is to decompose the uploaded model into two separate components: the malicious triggers and useful knowledge. By processing these components separately, we can *"take the essence and discard the dross"*, thus boosting the performance of FL. Our insight is that the backdoor only occupies a small part of knowledge of the model, as its key objective is to introduce a subtle perturbation to mislead specific results without affecting other execution logic. Thus, processing this part of knowledge, rather than the entire model, is intuitively beneficial to the final performance. As shown in Figure 1, our approach differs from traditional backdoor defenses, as we aim to extract the clean knowledge of the backdoored model for later aggregation. By doing so, we can maximize the utilization of benign information while mitigating the influence of the backdoor.

However, it is hard to directly identify the weights with malicious or useful knowledge in an uploaded model. To address this challenge, we propose a novel approach called **FedSKU**, which enables fine-grained backdoor removal through selective knowledge unlearning. Drawing inspiration from machine unlearning (Cao & Yang, 2015), our approach targets at *unlearning the malicious triggers while preserving the useful knowledge to be aggregated for each backdoored model.* In this way, we avoid the need to directly distinguish the model weights, while accurately removing the backdoor without sacrificing any other benign knowledge embedded in the model parameters.

To accomplish selective knowledge unlearning, FedSKU introduces several key techniques. **First**, we present a *pre-aggregation based* trigger recovery scheme, which is specially designed for FL to save the computational cost as well as providing the ingredient for the subsequent unlearning process. **Second**, to ensure that the unlearned model can still be effectively aggregated, we construct a surrogate model with the same dimensions as the uploaded model and conduct a *dual distillation* process to selectively transfer the knowledge into it. Here we design a novel distillation loss to enforce that only the clean knowledge is preserved in the surrogate model. Furthermore, to avoid the negative aggregation caused by the potential weight mismatch between the surrogate model and other benign models, we use the global model from the previous round as the initialization before distillation. This ensure that the weight divergence of the surrogate model is not too severe.

It is worth noting that, similar to current FL backdoor defenses, a detection step is still required to identify the malicious model. However, the main difference lies in that we further explore the possibility of the fine-grained utilization for this model. Therefore, FedSKU exhibits potential as a general module that can be easily integrated with existing detection-based defense mechanisms to further improve the performance of FL.

To validate the efficacy of our proposed approach, we conducted extensive experiments using public datasets that are widely employed for evaluating the FL backdoor defense performance. Our results demonstrate that, on top of the current FL defense methods, FedSKU can further achieve improved accuracy by up to 6.1% with a negligible ASR (Attack Success Rate) increase (<0.01%). Furthermore, we observe that FedSKU can significantly lower ASR compared to defense methods that simply extend knowledge distillation or machine unlearning techniques to the FL scenario. In-depth empirical analyses are also conducted, demonstrating the effectiveness of our framework.

The contributions of this paper are as follows:

- We propose a fine-grained backdoor defense method, called FedSKU, where we identify the malicious backdoor inside the uploaded models and selectively unlearn them. To the best

of our knowledge, this is the first attempt in the literature to study and explore the internal information of the uploaded models for improved FL performance.

- We design and develop a series of techniques to make the unlearning process more efficient and effective to benefit the FL process. As a result, we are able to generate an improved and clean federated global model for secure deployment.

- Extensive experiments on various datasets and attack modes demonstrate the superiority of FedSKU.

## 2 RELATED WORK

### 2.1 BACKDOOR ATTACK AND DEFENSE IN FL

Backdoor attack is a type of attack that involves manipulating a DNN model to behave maliciously in the presence of a specific trigger pattern (Zhong et al., 2020; Liu et al., 2020). In the field of FL, directly applying traditional backdoor attacks is infeasible since the injected malicious triggers are likely to be diminished when conducting the federated aggregation in the server side (Tolpegin et al., 2020). Under this condition, many researchers have proposed customized backdoor attacks that are specially designed for FL (Bagdasaryan et al., 2020; Zhou et al., 2021; Bhagoji et al., 2019; Sun et al., 2019). For instance, Bagdasaryan et al.(Bagdasaryan et al., 2020) first came up with the idea of model replacement to show how to backdoor federated learning, where attackers constrain and scale the malicious models based on the global model to dominate the aggregation process, thus deceiving the server. DBA (Xie et al., 2020) further introduced distributed backdoor attacks, breaking down the target trigger into multiple local triggers and assigning them to compromised participants. Specifically, each adversarial party utilized its own local trigger to corrupt the training data and then transmitted the poisoned update to the server, leading to a high attack success rate.

To cope with such attacks, several FL defense mechanisms have been proposed (?Cao et al., 2019; Nguyen et al., 2022). Generally, there are two steps in the process of FL backdoor defense. The first step is detecting the trigger added by the attacker and accurately locating uploaded models that have been backdoored. For example, in (Sattler et al., 2020), model updates are divided into clusters based on the cosine distance. In (Preuveneers et al., 2018), an unsupervised deep learning anomaly detection system is integrated into a blockchain process. The next step is to clean up the detected backdoor. Blanchard et al.(Blanchard et al., 2017) suggested a method based on the Krum function, which selected the optimal parameter vector to alleviate the effect of the malicious knowledge. FLAME (Nguyen et al., 2022) mitigated the toxicity by clipping and noising uploaded models. However, all the existing defense methods overlook or damage the benign knowledge embedded in the backdoored models, which significantly decreases the accuracy of the global model generated by the FL process.

### 2.2 MACHINE UNLEARNING

The term machine unlearning is originally proposed by Cao and Yang (Cao & Yang, 2015), where they presented an unlearning algorithm by transforming the learning into a summation form. Recently machine unlearning has been widely used in many areas (Du et al., 2019; Liu et al., 2021; Wu et al., 2022). Towards FL, Liu et al. (Liu et al., 2021) studied the unlearning problem in federated learning scenarios, where they adjusted the historical parameter updates of federated clients through the retraining process and the reconstruction of the unlearning model. Wu et al. (Wu et al., 2022) considered eliminating the client's contribution by subtracting the accumulated history updates from the model and restoring the model's performance using knowledge distillation methods. However, most of them have no relation to the backdoor defense. In the context of backdoor defense, BAERASER (Liu et al., 2022) was proposed to use the maximum entropy to recover trigger patterns and gradient-based forgetting, which strengthens harmless data to reverse the backdoor injection. NAD (Li et al., 2021) utilized a teacher network to guide the fine-tuning of the backdoored student network on a small clean subset of data such that the intermediate-layer attention of the student network can be aligned with that of the teacher network. Different from these unlearning methods that fail to achieve good defense performance when extending to the FL scenario (see Table 1), FedSKU presents a series of optimizations designed for the FL scenario, contributing to a more robust federated global model.

## 3 PROBLEM FORMULATION

### 3.1 BACKDOOR FORMULATION IN FL

Backdoor attacks have been widely studied for a single DNN model, where an attacker attempts to manipulate the DNN by introducing some triggers during the training pipeline. Unlike the traditional backdoor, in FL, the primary goal is to mislead the aggregation process since the final output of FL is a global model. In other words, poisoned local models generated by malicious clients must have a significant influence on the federation, such that they can effectively compromise FL.

Formally, assuming there are $N$ local clients, each of which contains a private dataset $D_i \in D = \{D_1, D_2, ..., D_N\}$. Assuming the client $N_{att}$ is a malicious user who wants to poison the global model $G$. Specifically, the attacker makes the global model behave normally on all input data except for specific attacker-chosen data $x \in trigger\_set$ for which attacker-guided incorrect predictions will be output. Here the trigger may be introduced to several data samples or the all set in $D_{att}$ and it can be implemented with different attack modes, such as flipping data labels or scaling up the weights of malicious models. Besides, due to the possibility of the huge number of clients involved in an FL system, some of them may establish collusion to collaboratively construct a backdoor, where each client holds a piece of the trigger (Xie et al., 2020) to make the attack.

### 3.2 SELECTIVE KNOWLEDGE UNLEARNING

In this paper, we introduce the idea of selective knowledge unlearning to defend the backdoor attacks in FL. Our approach, FedSKU, can simultaneously satisfy the following defender goals: (1) *Low attack success rate.* Because FedSKU explores the internal information of each backdoored model, the detection of malicious behaviors can be more precise so that we can effectively erase the backdoor. (2) *High final task performance.* Compared to traditional FL defense methods, FedSKU further takes advantage of the useful knowledge embedded in the poisoned local model, thus benefiting the final task performance since more knowledge is involved into the aggregation process.

Formally, given a series of local models $M = \{M_1, M_2, ..., M_N\}$ uploaded from client sides, we first identify the malicious ones $M_{att} = \{M_{att1}, M_{att2}, ...\}$ and further dive into their fine-grained information, decomposing each model into triggers $M_{att}^{tri}$ and useful knowledge $M_{att}^{use}$. As a result, we can selectively unlearn the triggers while preserving the useful information, which contributes to the subsequent aggregation. Based on the symbols, we define the goal of our *Selective Knowledge Unlearning* as follows.

**Definition 3.1. (Selective Knowledge Unlearning)**. *Let $GACC$ and $ASR$ be the final task performance of the global model and the attack success rate, respectively. When conducting FL, the goal of SKU is to selectively unlearn $M_{att}^{tri}$ and generate a clean model $M_{att}^{cle}$ for aggregation, such that we can obtain an improved global model $G_{pro}$ with high $GACC$ and low $ASR$.*

## 4 METHOD

### 4.1 OVERVIEW

We design and implement FedSKU, a framework to achieve fine-grained backdoor removal via selective knowledge unlearning. Figure 2 depicts the overall pipeline of FedSKU, which can be briefly summarized as follows. First, we follow the traditional methods to pick out the backdoored model (e.g., anomaly detection part of FLAME (Blanchard et al., 2017) or Krum (Nguyen et al., 2022)), which is then processed by a *trigger recovery* module and a *trigger unlearning* module. In the *trigger recovery* module, we take advantage of the backdoored model and a few public data to separately recover the trigger pattern. During the process, a pre-aggregation scheme is proposed to ensure efficiency. In terms of this pattern, we propose a novel *trigger unlearning* method to accurately unlearn the specific triggers while selectively transferring the useful knowledge into a surrogate clean model. In this way, the generated surrogate model is able to contribute to the aggregation process for an improved global model. In the remainder of this section, we describe in detail our approach for implementing the two key modules.

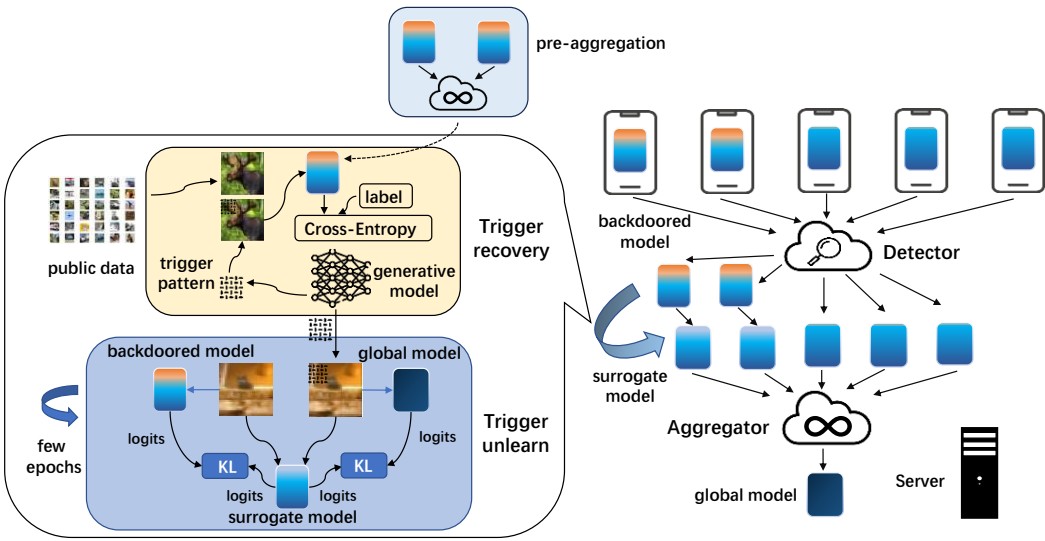

Figure 2: Overview of the proposed FedSKU framework.

## 4.2 TRIGGER RECOVERY

Given a backdoored model, we first need to recover the trigger for the later unlearning process. Instead of trying to recover the original trigger, we design a novel *pre-aggregation based recovery scheme* to efficiently get a valid trigger distribution, with the help of MESA(Qiao et al., 2019).

Specifically, the key idea of MESA is to approximate generator G by training N sub-models $G_1, G_2, ..., G_N$, and each sub-model $G_i$ only learns a part of trigger distribution. The sub-model $G_i$ can be updated through the loss function $L$:

$$L = \frac{1}{l} \sum_{x \in D_{pub}} \left( \max \left( 0, \gamma_i - M_{att}(x + G_{\theta_i}(z)) \right) - \lambda H \left( G_{\theta_i}(z); z' \right) \right). \tag{1}$$

where $D_{pub}$ is the public dataset containing $l$ samples and $x$ represents a sample from the public dataset. $z$ and $z'$ are independently extracted from a normal distribution with a mean of 0 and a standard deviation of 1 for mutual information (MI) estimation. $H \left( G_{\theta_i}(z); z' \right)$ defines the entropy and is equivalent to its MI. $\theta$ is defined as the parameters of the sub-model $G_{\theta_i}$. $\gamma$ is the threshold and $\lambda$ balances the constraint and the entropy. In this way, we can generate the trigger distribution as the attack pattern.

In FL, there may exist a huge number of clients in an FL system, indicating that the number of attackers is also unnegligible. Under this condition, directly applying the above mentioned MESA may introduce considerable training overhead since we require generating the trigger pattern for each backdoored model. FedSKU addresses the issue by introducing a *pre-aggregation* scheme to the backdoored models before conducting the trigger recovery. Our insight is that the main objective of colluded attackers is to mislead the aggregation process and if we pretend to aggregate, the generated model will definitely hold the specific malicious features. As a result, instead of conducting the recovery process for each backdoored model, we only need to process the pre-aggregated model, getting rid of the massive computational consumption.

## 4.3 TRIGGER UNLEARNING

After obtaining the trigger pattern, we next utilize it to conduct our selective knowledge unlearning. Concretely, we design a *dual distillation* method to achieve our goal. In the following part, we elaborate our key designs: distillation architecture and distillation loss. Note that the distillation process require some public data and this setting is widely accepted in the field of FL (Lin et al., 2020; Cho et al., 2022).

**Distillation architecture.** Considering that the distilled model requires participating in the later aggregation process, we first construct a surrogate model with an identical dimension to each backdoored model as the distillation student, such that they can be directly federated with other benign models. Here we adopt a *dual distillation* pipeline with two different teachers for selective knowledge unlearning and transfer.

Specifically, the first teacher is the backdoored model, where we only use the clean data as the input to distill the useful knowledge and transfer it to the surrogate model. Note that the malicious knowledge will not disturb this distillation process since our input has no relationship with respect to the trigger. Besides, to further isolate that the trigger logic is indeed isolated from the surrogate model, we design another clean teacher and use the data with introduced trigger patterns as the input. In this way, the feature information of the trigger can be further diminished by the distillation process.

Directly conducting such a distillation pipeline seems effective in defending backdoor attacks. However, in the context of FL, we should additionally take the subsequent aggregation process into account because the final goal of FL is generating a better global model. Here we observe that distillation may lead to the weight mismatch issue as the learning degree between the surrogate model and other benign models can be significantly different. To address the problem, we resort to the global model from the previous round as the initialization of the surrogate model. The intuition behind this design is that the previous global model can provide more generalized clean knowledge and better-aligned model weights, thereby simultaneously facilitating the process of knowledge transfer and enhancing the effectiveness of subsequent aggregation.

**Distillation loss.** The key principle of the distillation loss is to block the transfer of backdoored knowledge to the surrogate model and enable the useful knowledge flowing to it. As described in the last part, two teachers are constructed to accomplish our goal.

Formally, supposing that the surrogate model is $S(x)$. The global model of the previous round is $T(x)$ and the backdoored model is $M_{att}(x)$. Their output logits are represented by $s, t, m$, respectively. We achieve the goal of extracting the useful knowledge from the global model by defining KL-Divergence between $T(x)$ and $S(x)$ based on the poisoned data, which is formulated as:

$$\mathcal{KL}\left(T(x) \| S(x)\right) = \sum_{d_i \in D_{pub}} t^{(d_i + G(z))} \log\left(t^{(d_i + G(z))} / s^{(d_i + G(z))}\right) \tag{2}$$

where the public dataset is denoted as $D_{pub}$, $z$ is the random noise defined by $z \sim \mathcal{N}(0, 1)$ and $G(z)$ is the generative model of the trigger distribution described in 4.2. Here we denote the poisoned data as $d_i + G(z)$, where $d_i$ is the clean data and $G(z)$ is the generated trigger pattern. Based on Eq. 2, we are able to enforce $S(x)$ to perform like $T(x)$ under the poisoned data, which potentially ensures the surrogate model discarding the malicious knowledge from the $M_{att}(x)$ since $T(x)$ can be considered as a clean model.

In addition, we also want $S(x)$ to absorb the useful knowledge from $M_{att}(x)$. Therefore, we define the KL-Divergence between $S(x)$ and $M_{att}(x)$ to accurately extract the useful knowledge, with the help of a few clean data. This process can be denoted as:

$$\mathcal{KL}\left(M_{att}(x) \| S(x)\right) = \sum_{d_i \in D_{pub}} m^{(d_i)} \log\left(m^{(d_i)} / s^{(d_i)}\right) \tag{3}$$

In Eq equation 3, we enforce $S(x)$ to perform similarly to $M_{att}(x)$. As the malicious model $M_{att}(x)$ is triggered only when exposed to meticulously designed data and performs normally as benign models in other situations, the knowledge of the malicious model could be utilized by using clean data as the input.

Finally, the overall unlearning objective can be formulated as:

$$L_{distill}(x) = \mathcal{KL}\left(T(x) \| S(x)\right) + \beta * \mathcal{KL}\left(M_{att}(x) \| S(x)\right) \tag{4}$$

where $\beta$ is a hyperparameter that balances the two distillation processes.

In this way, FedSKU transforms the backdoored models to a series of clean surrogate models, which can be used to participate in the aggregation process for improved performance.

Table 1: Results on different datasets with two typical FL backdoor attacks. Note that FLAME uses a different backbone model (ResNet-18) compared to other baselines (WideResNet) and we follow each setting to generate corresponding results.

| Method | CIFAR-10 | | | | CIFAR-100 | | | |
| --- | --- | --- | --- | --- | --- | --- | --- | --- |
| | constrain-scale | | DBA | | constrain-scale | | DBA | |
| | ASR | GACC | ASR | GACC | ASR | GACC | ASR | GACC |
| *FLAME* | 2.97% | 73.59% | 3.30% | 71.48% | 0.28% | 57.44% | 0.31% | 55.33% |
| *FLAME+Ours* | 3.35% | **75.40%** | 3.61% | **72.49%** | 0.37% | **63.42%** | 0.32% | **61.43%** |
| *BAERASER* | 14.16% | 68.72% | 30.57% | 70.22% | 0.87% | 49.98% | 4.22% | 51.88% |
| *NAD* | 34.61% | 68.17% | 34.71% | 68.34% | 1.86% | 41.82% | 75.10% | 51.30% |
| *Ours* | **2.55%** | 67.61% | **2.51%** | 68.62% | **0.55%** | 51.01% | **0.48%** | 48.28% |

## 5 EVALUATION

### 5.1 EXPERIMENTAL SETUP

**Backdoor settings.** We implement two typical backdoor attacks, *Constrain-and-scale* (Bagdasaryan et al., 2020) and *DBA* (Xie et al., 2020), which are specially designed for FL. Besides, we also employ *Badnets* (Gu et al., 2017) for the baselines that fail to defend the above two attacks. For a fair evaluation, we follow the configuration, including the trigger patterns, trigger sizes and the target labels, of these attacks in their original papers. We test the performance of all attacks on three benchmark datasets, CIFAR-10/CIFAR-100 (Krizhevsky et al., 2009) and Tiny-Imagenet (Le & Yang, 2015), with ResNet-18 (He et al., 2016) and WideResNet (WRN-16-1*) (Zagoruyko & Komodakis, 2016) being the base models throughout the experiments. More details on attack configurations and implementations can be found in the appendix.

**FedSKU settings.** As for the trigger recovery process, FedSKU employs the same trigger recovery technique with BAERASER(Liu et al., 2022), and our setting of trigger recovery follows BAERASER. Additionally, the trigger recovery threshold is set to be 0.55 for the CIFAR10 and CIFAR100 datasets and 0.4 for the Tiny-imagenet dataset. Besides, we set $\beta$ to 10 and the number of training epochs to 4. Additionally, we set the distillation temperature to 1. For FedSKU, Baeraser, and NAD, we randomly sample 5% of the data from the test set to obtain the training data required for distillation and unlearning. These sampled data points are separated from the test set and are not used during testing, which is also consistent with the source code of BAERASER.

**Baselines.** We compare FedSKU with two state-of-the-art defense methods for FL, Krum (Blanchard et al., 2017) and FLAME (Nguyen et al., 2022), which respectively represent the *removal-based* and *noise-based* defense. Considering that our approach is orthogonal to these methods, the performance is evaluated by incorporating FedSKU into them. In addition, to validate the effectiveness of our selective knowledge unlearning, we compare two representative distillation and unlearning defense schemes, *NAD* (Li et al., 2021) and *BAERASE* (Liu et al., 2022), which are extended to the FL scenario. We provide more details on the defense baselines in the appendix.

**Evaluation metrics.** To assess the effectiveness of defense mechanisms, we utilize two metrics: the attack success rate (ASR) and the global model accuracy (GACC), which is evaluated after the aggregation process. ASR represents the proportion of backdoored examples that are wrongly classified as the intended label. Meanwhile, GACC measures the accuracy of the global model on uncontaminated samples. The strength of a defense mechanism is indicated by a significant reduction in ASR and a high performance in GACC.

### 5.2 PERFORMANCE COMPARISON

**GACC and ASR comparison.** In this part, we report the GACC and ASR performance of different methods. Here we only record the performance on CIFAR-10 and CIFAR-100 due to the limited pages. Results for Tiny-Imagenet can be found in the appendix. Note that Krum can only defend *Badnets*, which means that the effectiveness of FedSKU can only be reflected on that attack mode. Table 2 illustrates the results. We can clearly see that by incorporating FedSKU, the final accuracy of

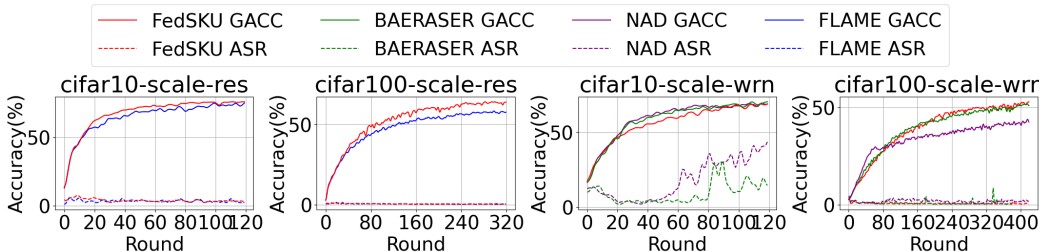

Figure 3: Convergence performance on different methods.

the global model can be enhanced with a slight increase in ASR, which validates the benefit of our framework in adding useful knowledge into the aggregation process.

Table 1 exhibits the performance of other baselines and ours under the two typical FL attacks. From the table, we can observe that: (1) Compared to *FLAME*, the proposed approach can achieve the GACC improvement by up to 6.1%, while only incurring neglectable ASR increase (<1%). This demonstrates that FedSKU indeed introduces more useful knowledge to benefit the federation process. (2) Although other unlearning or distillation baselines show the superiority in defending backdoor attacks in a single DNN, when extending to FL, their effectiveness has a significant decrease, which is reflected by the high ASR. Different from them, FedSKU introduces a series of techniques that are specially designed for FL, thus making the global model more robust.

**Convergence comparison.** We record the GACC and ASR of each round in FL and plot the convergence lines of different methods. As illustrated in Figure 3, we visualize the training state under the *constrain and scale* attack. We can find that the performance improvement of our method is significantly better on CIFAR-100 compared to CIFAR-10. This may be due to the limited data in cifar10, making it more prone to overfitting during the local learning, which subsequently affects the quality of aggregation. Additionally, in CIFAR-10, many baselines exhibit large fluctuations in ASR, indicating that FL backdoors are even more challenging to defend if we only have a small dataset. However, FedSKU, due to its fine-grained consideration of the internal model information, is not affected by the data volume.

Table 2: Results of different datasets on Badnets.

| Method | Badnets | | | |
|---|---|---|---|---|
| | CIFAR-10 | | CIFAR-100 | |
| | *ASR* | *GACC* | *ASR* | *GACC* |
| Krum | 3.05% | 63.58% | 0.36% | 47.25% |
| Krum+Ours | 4.03% | 66.50% | 0.64% | 49.85% |

**Ablation study.** In our design, we employ the global model of the previous round as the initialization of the surrogate model, in order to alleviate the problem of weight mismatch. Here we explore whether this scheme is effective by replacing it with a randomly initialized model and a backdoored model as the teacher. As shown in Table 3, we test the ASR and GACC on the CIFAR-10 datasets with two attack modes. We can draw the following conclusions from the table: (1) Although using the backdoored model as the teacher can achieve remarkable performance on GACC, the backdoor is also embedded into the global model with a high ASR, suggesting that we fail to defend the attacks. (2) A randomly initialized model can effectively mitigate backdoor attacks; however, the defense comes at the cost of significantly reducing the overall accuracy of the global model. We believe this is due to the issue of weight mismatching during the aggregation process. In contrast, by utilizing the global model of the last round to align the weights between the student model and other benign models, we can ensure higher accuracy while still defending against attacks.

Table 3: Effect of different components to be the initialization of the surrogate model.

| Teacher | constrain-and-scale | | DBA | |
|---|---|---|---|---|
| | *ASR* | *GACC* | *ASR* | *GACC* |
| *random model* | 1.79% | 66.55% | 3.19% | 71.32% |
| *backdoored model* | 90.86% | 75.35% | 97.94% | 75.55% |
| *global model* | 2.76% | 75.61% | 2.91% | 71.99% |

Table 4: Impact of the non-iid degree on CIFAR-10.

| Method | 0.2 | | 0.4 | | 0.6 | | 0.8 | |
|---|---|---|---|---|---|---|---|---|
| | *ASR* | *GACC* | *ASR* | *GACC* | *ASR* | *GACC* | *ASR* | *GACC* |
| *BAERASER* | 6.97% | 69.01% | 14.22% | 65.27% | 9.79% | 51.14% | 13.16% | 33.88% |
| *NAD* | 15.75% | 68.80% | 18.41% | 64.55% | 30.97% | 52.98% | 16.24% | 32.53% |
| *FLAME* | 2.66% | 67.54% | 3.60% | 58.94% | 4.74% | 44.93% | 11.82% | 22.83% |
| *Ours* | 2.96% | 68.12% | 2.84% | 62.44% | 4.20% | 49.45% | 12.04% | 25.44% |

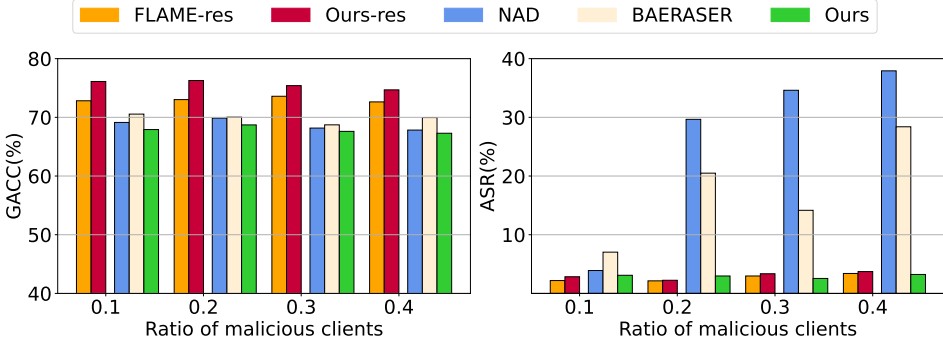

Figure 4: Impact of the ratio of malicious clients.

## 5.3 IMPACT OF THE NON-IID DEGREE

In the default experimental settings, we assume that the data in each client follow an independent identically distributed (iid) situation. However, in real-world scenarios, data are usually non-iid due to the various environments of different users. This subsection studies the impact of the non-iid degree on different methods. Specifically, we follow the non-iid setting in (Nguyen et al., 2022) to conduct the experiments on CIFAR-10. Table 4 demonstrates the results, where GACC and ASR are respectively recorded to evaluate the performance. From the figure, we can see that as the non-iid degree increases, the GACC of all methods degrades dramatically. However, FedSKU can maintain a low ASR compared to others. Although FLAME also can achieve a comparable ASR performance to ours, its GACC performs worse, especially for the high non-iid degree. However, when the degree reaches 0.8, both ASR and GACC are largely affected for all methods, which means that existing methods cannot cope with an extreme non-iid situation.

## 5.4 IMPACT OF THE RATIO OF MALICIOUS CLIENTS

The ratio of malicious clients plays an important role in the GACC since it directly determines the amount of useful knowledge in FL. In our default settings, we assume there are 30% malicious clients involved in an FL system. Here we manually set the malicious ratio to 0.1-0.4 to study the impact. As illustrated in Figure 4, we record the performance of different ratios under the *constrain and scale* attack mode on CIFAR-10. We can find that compared to FLAME, FedSKU consistently achieves better GACC performance, with only a marginal increase in ASR, regardless of the ratio of malicious clients. Besides, in contrast to other baselines, as the ratio of malicious clients increases, we are able to maintain the ASR at a very low value, further indicating the robustness of FedSKU.

## 6 CONCLUSION

In this paper, we defend the backdoor attack in the context of FL with a fine-grained approach. We demonstrate our solution through FedSKU, a novel framework to achieve backdoor defense with the help of selective knowledge unlearning. Concretely, we selectively unlearn the malicious triggers while preserving the useful knowledge to be aggregated, which not only mitigates the backdoor trigger but also enhances the performance of the final global model since more useful knowledge is involved in the aggregation phase. Extensive experiments demonstrate the effectiveness of FedSKU, significantly outperforming other state-of-the-arts.

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

Table 5: Statistics of the benchmark datasets and parameters used for implementing different attacks. Here "Be" denotes Benign and "Po" denotes Poison. "Num" represents *num_clients*.

| Dataset | #Training | #Testing | Attack | Num/Frac | Model | Be(Ib/Eb) | Po(Ip/Ep) |
|---|---|---|---|---|---|---|---|
| CIFAR-10 | 50000 | 10000 | *constrain* | 100/0.1 | ResNet | 0.1/2 | 0.05/2 |
| | | | *DBA* | 100/0.16 | | 0.1/2 | 0.05/6 |
| | | | *Badnet* | 100/0.1 | | 0.1/2 | 0.05/6 |
| | | | *constrain* | 100/0.1 | WideRes | 0.1/2 | 0.05/8 |
| | | | *DBA* | 100/0.16 | | 0.1/2 | 0.05/8 |
| CIFAR-100 | 50000 | 10000 | *constrain* | 100/0.1 | ResNet | 0.1/3 | 0.05/8 |
| | | | *DBA* | 100/0.16 | | 0.1/2 | 0.05/6 |
| | | | *Badnet* | 100/0.1 | | 0.1/2 | 0.05/8 |
| | | | *constrain* | 100/0.1 | WideRes | 0.1/3 | 0.05/8 |
| | | | *DBA* | 100/0.16 | | | |
| TinyImageNet | 100000 | 10000 | *constrain* | 100/0.1 | ResNet | 0.1/3 | 0.05/12 |
| | | | *DBA* | 100/0.16 | | | |

# A    APPENDIX

# B    DETAILED EXPERIMENTAL SETTINGS

## B.1    BENCHMARK DATASETS

In our experiments, we evaluate baselines and our method against *constrain-and-scale* and *DBA* on three standard benchmark dataset: CIFAR-10 (Krizhevsky et al., 2009), CIFAER-100 (Krizhevsky et al., 2009) and Tiny-imagenet (Le & Yang, 2015). The detailed statistics are illustrated in Table 5 and we briefly introduce each dataset as follows.

- *CIFAR-10 (Krizhevsky et al., 2009)*: CIFAR-10 consists of 60000 32x32 color images in 10 classes and each of which has 6000 images including 5000 training images and 1000 test images.

- *CIFAR-100 (Krizhevsky et al., 2009)*: The CIFAR-100 dataset is a classification dataset consisting of 60000 32x32 color images with 100 classes and 600 images in each class. Each class has 500 training images and 100 testing images. CIFAR-100's 100 classes are divided into 20 superclasses. Each image has a "fine-grained" label (the class it belongs to) and a "coarse-grained" label (the superclass it belongs to).

- *Tiny-ImageNet (Le & Yang, 2015)*: Tiny-ImageNet is a subset of the ILSVRC2012 classification dataset consisting of 100000 images downsized to 64x64 colour images. It contains 200 classes and each class has 500 training images, 50 validation images, and 50 test images.

## B.2    BASELINES AND HYPERPARAMETER SETTINGS

FedSKU is incorporated into Krum and FLAME to observe the performance. Besides, on top of FLAME, FedSKU is compared to BAERASER+FLAME and NAD+FLAME to validate the effectiveness. We briefly summarize these methods as follows:

- *Krum (Blanchard et al., 2017):* The Krum algorithm is a robust aggregation technique designed to mitigate the effect of malicious or faulty participants in a distributed system. It aims to select a trustworthy model by considering the agreement among a subset of the most reliable models. We re-implemented Krum according to the original paper.

- *FLAME (Nguyen et al., 2022):* FLAME is a defense framework including 3 key components: filtering, clipping, and noising. They first use a clustering-based approach to identify and remove malicious model updates with a high attack impact, then clip the model to limit the impact of scaled models. In the end, they calculate the adequate level of noise required to

guarantee the eradication of backdoors. We re-implemented FLAME based on (Nguyen et al., 2022).

- *FLAME+BAERASER (Liu et al., 2022):* BAERASER utilizes MESA (Qiao et al., 2019), a generative model to recover the trigger distribution and then uses it to conduct machine unlearning. Concretely, BAERASER designs a *gradient ascent* based approach to achieve machine unlearning. We extend BAERASER to the FL scenario on the top of FLAME.

- *FLAME+NAD (Li et al., 2021):* NAD utilizes a distillation process to erase the backdoor triggers. In NAD, they first fine-tune the original backdoored model to obtain a teacher model. This teacher model is then used to perform distillation, aiming to align the backdoored student model's intermediate-layer attention with those of the teacher model. Similar to BAERASER, we extend NAD to the FL scenario on the top of FLAME.

**Backdoor setting.** Towards the *constrain-and-scale* attack and DBA attack, we did not incorporate a scaling step since this would make the model unavailable or ineffective. On the other hand, the scaling model is actually easy to be detected by current anomaly detection-based methods (Nguyen et al., 2022). But we still follow (Xie et al., 2020) and (Bagdasaryan et al., 2020) to train the backdoored model with an anomaly detection term that measures the L2 norm between the backdoored model with the global model.

We have $num\_clients$ clients in total and distribute all the training images to them in the beginning. Following (Nguyen et al., 2022), we vary the degree of non-iid data $q$ by altering the proportion of images provided to clients that belong to a specific class. At each global training epoch, we randomly choose $frac$ of $num\_clients$ clients to participate in this round of training. Each benign client trains its local model using SGD optimizer with local learning rate $l_b$ for $E_b$ local epochs and the batch size is 128. Each malicious client trains its local model with local learning rate $l_p$ for $E_p$ local epochs and the batch size is 128. For detailed parameter settings, please refer to Table 5.

We have observed that the effectiveness of unlearning the models obtained during the initial rounds of federated aggregation is quite poor. At this stage, the models have not yet reached enough stability, and attempting to unlearn them would be inappropriate. Therefore, we allow all the unlearning-based methods to unlearn the malicious models after $E_{start}$ epochs. And for all the experiments, we set $E_{start}$ as 15.

**Baseline setting.** To validate the effectiveness of our selective knowledge unlearning, we also incorporate BAERASER and NAD into FLAME to extend them to the FL scenario. We use the source code provided by (Liu et al., 2022) and (Li et al., 2021) to implement the defense. For NAD, we evaluate its performance only with WideResNet as the paper (Li et al., 2021) did. Directly extending NAD to the FL scenario based on the source code of NAD would introduce several challenges. The primary concern is the mismatch of learning rates and the varying learning rate requirements for the unlearning task across different global training epochs. However, since these challenges are not encountered in regular scenarios, we have made the following adjustments: Firstly, we obtain the initial learning rate of the fine-tuned teacher model as 0.01, and we decrease it by half every 40 global training cycles until it reaches 0.0001. Secondly, the initial learning rate for distillation is set to 0.01. The remaining settings remain consistent with the source code of NAD.

We also introduce the learning rate adjustments to our extension of BAERASER to the FL scenario. We set the initial learning rate to 5e-5 and similarly decrease it by half in every 40 global training cycles until it reaches 5e-6. The remaining settings remain unchanged according to the source code of BAERASER.

**Implementation details.** All our experiments are simulated and conducted in a server that has 3 GeForce GTX 3090 GPUs, 48 Intel Xeon CPUs, and 128GB memory. We implement FedSKU in Python with PyTorch. Similar to the settings in *NAD* (Li et al., 2021), we assume all defense methods have access to the same 5% of the clean training data for distillation or unlearning. Besides, we run the experiments 3 times and reported the average results.

## C  ADDITIONAL EMPIRICAL RESULTS

Due to the page limitation of the main text, here we show our additional empirical results to further demonstrate the superiority of FedSKU.

Table 6: Results on the TinyImageNet dataset.

| Method | constrain-and-scale | | DBA | |
|---|---|---|---|---|
| | ASR | GACC | ASR | GACC |
| FLAME | 0.76% | 29.95% | 0.46% | 31.41% |
| BAERASER | 0.73% | 35.39% | 27.61% | 34.94% |
| Ours | 0.65% | 35.27% | 0.56% | 33.69% |

Table 7: Effect of different components to be the initialization of the surrogate model on CIFAR-100.

| Teacher | constrain-and-scale | | DBA | |
|---|---|---|---|---|
| | ASR | GACC | ASR | GACC |
| random model | 0.50% | 49.98% | 0.31% | 56.22% |
| backdoored model | 90.70% | 63.07% | 97.39% | 64.13% |
| global model | 0.28% | 65.08% | 0.27% | 63.36% |

## C.1 MAIN PERFORMANCE ON THE TINYIMAGENET DATASET

We also conduct experiments on the Tiny-ImageNet dataset and record the results in Table 6. During our experiments, we find that training WideResNet on the Tiny-ImageNet dataset is challenging, with significantly slow and poor convergence. Furthermore, it is important to note that the parameters and settings provided in the paper regarding intermediate-layer attention is specific to WideResNet. This poses several challenges when attempting to transfer NAD to ResNet18 due to the parameter adjustment and configuration discrepancies. Therefore, we only use ResNet18 for the Tiny-ImageNet dataset and do not include NAD in the experiments.

As shown in Table 6, only FedSKU and FLAME could maintain low ASR under the two attack modes. On the other hand, FedSKU always gains a better GACC compared with FLAME under two attack modes. Along with the results on CIFAR-10 and CIFAR-100 datasets, it shows the superiority of our method under various datasets.

## C.2 ABLATION STUDY ON OTHER DATASETS

Table 8: Effect of different components to be the initialization of the surrogate model on TinyImageNet.

| Teacher | constrain-and-scale | | DBA | |
|---|---|---|---|---|
| | ASR | GACC | ASR | GACC |
| random model | 0.92% | 28.06% | 0.66% | 32.87% |
| backdoored model | 0.16% | 34.97% | 24.54% | 34.57% |
| global model | 0.75% | 35.38% | 0.60% | 33.32% |

We also explore the effectiveness of setting the global model of the previous round as the initialization of the surrogate model on the CIFAR-100 dataset and the Tiny-ImageNet dataset with two attack modes. In this case, we replace the global model with a randomly initialized model and a backdoored model for comparison. We can draw similar conclusions from the tables as before. Specifically, in Table 7 and Table 8, we can still observe that using a backdoored model easily embeds the backdoor into the global model on the CIFAR-100 dataset, as evidenced by the high ASR value, no matter what attack mode is. As for the results on the Tiny-Imagenet dataset, we can observe that using a backdoored model does not yield as high ASR as in CIFAR-100 or CIFAR-10. We believe this phenomenon can be attributed to two main factors. Firstly, training on the Tiny-Imagenet dataset

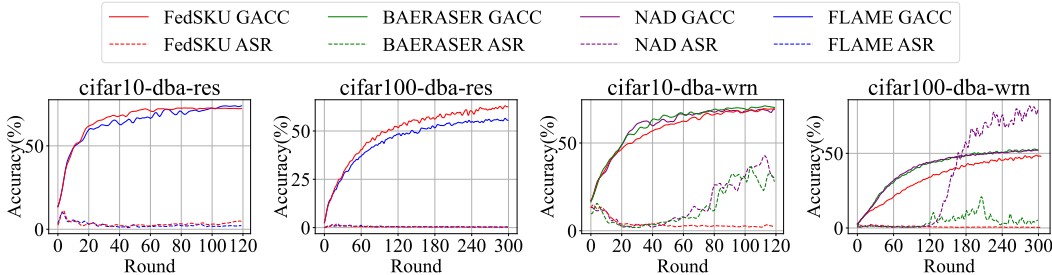

Figure 5: Convergence performance on different methods under the DBA attack.

Table 9: Impact of the non-iid degree on CIFAR-10 under the DBA attack.

| Method | 0.2 | | 0.4 | | 0.6 | |
|---|---|---|---|---|---|---|
| | *ASR* | *GACC* | *ASR* | *GACC* | *ASR* | *GACC* |
| *BAERASER* | 17.52% | 69.42% | 3.15% | 66.63% | 8.71% | 58.38% |
| *NAD* | 27.45% | 67.61% | 10.12% | 66.27% | 23.42% | 57.43% |
| *FLAME* | 2.52% | 67.44% | 1.93% | 63.05% | 1.34% | 49.99% |
| *Ours* | 2.73% | 67.02% | 2.39% | 63.77% | 1.34% | 52.30% |

is generally more challenging compared to CIFAR-100 and CIFAR-10, making it inherently more difficult to embed backdoors successfully. Secondly, FedAvg aggregates model updates from multiple participants, which helps in diluting the influence of the backdoored models.

On the other hand, using a randomly initialized model can effectively mitigate backdoor attacks, but it results in a significant loss in GACC. In contrast, utilizing the global model from the previous round as the initialization of the surrogate model proves to be effective in achieving high GACC while mitigating the impact of backdoors. The results obtained on both the CIFAR-100 and Tiny-Imagenet datasets further validate the effectiveness of this scheme.

### C.3 DETAILED ANALYSIS UNDER THE DBA ATTACK

**Convergence comparison.** Figure 5 shows the convergence performance of different methods under the DBA attack. We can observe that FedSKU can achieve improved accuracy with lower ASR. Besides, another interesting finding is that under this attack mode, NAD obtains a high ASR, significantly outperforming others. The reason may be that this distributed attack can easily bypass the distillation process used in NAD. However, FedSKU is not affected by it, suggesting the superiority of our distillation process.

**Impact of the non-iid degree.** We also conduct experiments on the impact of the non-iid degree in the context of the DBA attack, and the results are presented in Table 9. Similar to the results obtained by applying the constrain-and-scale attack, we can observe that as the non-iid degree increases, only FedSKU and FLAME maintain a relatively low ASR compared to the other methods. However, we observe that FLAME experiences a more significant decrease in GACC as the non-iid degree becomes higher and FedSKU performs better.

**Impact of the ratio of malicious clients.** As shown in Figure 6, FedSKU is still effective in improving the final accuracy on FLAME and lowering ASR regardless of the attack mode. Also, other baselines cannot achieve desirable performance under the DBA scenario, which further exhibits the superiority of the proposed approach.

## D   ALGORITHM AND REPRODUCTION

Algorithm 1 shows the whole pipeline of our proposed FedSKU. For the client side, we conduct local training for benign clients. Toward the attackers, we follow the typical attack pipelines to introduce

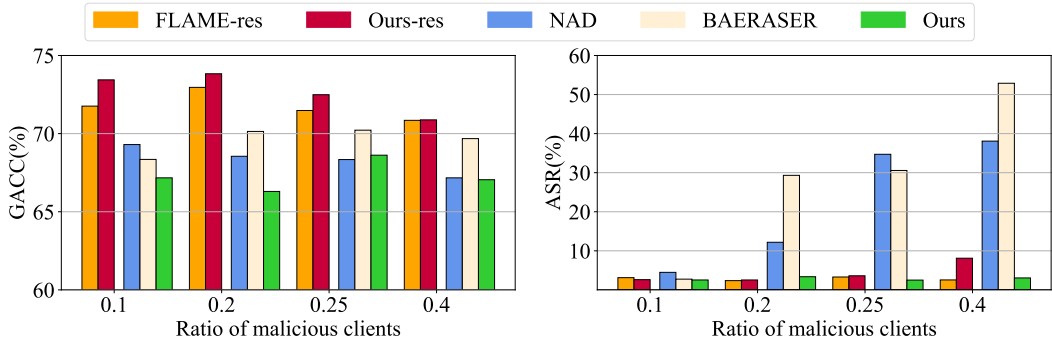

Figure 6: Impact of the ratio of malicious clients under the DBA attack.

---
**Algorithm 1** Pipeline of FedSKU
---
**Operation**:

1: Distribute a shared model $M_{ini}$ to each client
2: **for** $i=1$ to $R$ **do**
3:     Conduct these steps in **Client side** and **Server side** sequentially
4: **end for**
5: Obtain the final improved global model $G_{final}$

**Client side**:

1: For benign clients, train $M_{ini}$ with its own task dataset $D$ and generate clean local models $M_{clean}$; For attackers, introduce the backdoors by existing methods to generate $M_{att}$
2: Upload $M_{clean}$ and $M_{att}$ to the server
3: Wait for the predicted values sent back from the server

**Server side**:

1: Collect the uploaded Models $M_{clean}$ and $M_{att}$
2: Pick out $M_{att}$ with existing detection methods
3: Recover the trigger from $M_{att}$ based on Eq. 1
4: Conduct trigger unlearning based on Eq. 2, Eq. 3 and Eq. 4, generating a series of clean surrogate models $S$
5: Aggregate the clean local models $M_{clean}$ and surrogate models $S$ to obtain an improved and robust global model $G_{pro}$

---

the backdoors. Our defense primarily focuses on the server side, where we conduct our selective knowledge unlearning to achieve the mitigation of attacks as well as enhancing the final performance. Specifically, we recover the trigger from malicious clients and leverage it to only transfer the useful knowledge into corresponding surrogate models, which can be used to participate in the aggregation process for an improved global model. Note that the operations will be implemented sequentially until training convergence.

To ensure reproducibility, we have provided the overview of datasets and baselines in Appendix B.1 and Appendix B.2. Our experimental environment is presented in Section 5.1. We will make our code and other artifacts available to the community after the notification.

# E    DISCUSSIONS

## E.1    THREAT MODEL

In our paper, we follow the attack setting of the previous FL defense work (Nguyen et al., 2022). The attacker $\mathcal{A}$ intends to inject the backdoor pattern into the aggregated model, which will subsequently

affect all $N$ participants. The attacker controls $m < \frac{N}{2}$ clients and utilizes them to upload the elaborate model updates to the server. The attacker has full knowledge of the clients under his control, including the local training dataset, training processes, and model parameters. However, the attacker has no information about all other clients and no control of the server. The attacker intends to poison the global model $G$ and simultaneously makes it behave as if it is a normal model. Specifically, the attacker's goal is:

$$\forall x' \in T, G(x') = y_{\text{Att}} \tag{5}$$

and

$$\forall x_i \in D, G(x_i) = y_i. \tag{6}$$

Here, $y_{\text{Att}}$ is the attacker's target label, $y_i$ denotes the correct label of input $x_i$, $D$ is the clean data and $T$ denotes the data containing the backdoor pattern chosen by the attacker.

In addition, regarding the public data used in the trigger recovery and unlearning process, our method requires a small subset of clean data with the same data type as our training data. Consistent with the settings used in NAD (Li et al., 2021), we only utilize 5% data of the dataset as the public data.

As for the backdoor types, our method is effective against typical backdoor attacks that target a subset of inputs, meaning that the backdoor is activated only when the input data contains a specific trigger pattern.

### E.2 OVERHEAD

Our approach introduces an unlearning process compared to conventional removal-based defense, which comprises a trigger recovery part and an unlearning part. In the following parts, we analyze the overhead in detail.

**Trigger Recovery:** In the trigger recovery phase, we start by searching for the attacker's target label. We denote the label currently being searched for as $i$ and the number of the label is $num\_class$ (denoted as $N_c$). We then apply our recovery process to obtain multiple sub-models, denoted by $G_i$, where $i = 1, \ldots, N_{sub}$. For each sub-model, we train for a number of epochs defined by $num\_recovery\_epoch$ (abbreviated as $E_r$) according to Equation (1). Assuming the dataset size at the server is $V$ and the batch size during training is $recovery\_batch\_size$ (abbreviated as $BS_r$), the complexity of the recovery part can be described as performing $N_c \times E_r \times N_{sub} \times \frac{V}{BS_r}$ backpropagations.

**Unlearning:** After obtaining the final set of sub-models, $G = [G_1, G_2, \ldots, G_{N_{sub}}]$, we proceed to conduct unlearning (i.e., distillation). This involves conducting $distill\_epoch$ (abbreviated as $E_d$) training epochs with Equation (4) for unlearning. The batch size for distillation is denoted as $distill\_batch\_size$ (abbreviated as $BS_d$), leading to a distillation complexity of $E_d \times \frac{V}{BS_d}$ backpropagations.

**Total overhead:** Let us denote the number of detected anomalous clients as $abnormal\_client\_num$ (abbreviated as $N_a$). Taking into account the pre-aggregation strategy, the overall complexity of our approach for the detected anomalous clients can be expressed as follows:

$$\text{Total Overhead} = N_c \times E_r \times N_{sub} \times \frac{V}{BS_r} + N_a \times (E_d \times \frac{V}{BS_d}) \tag{7}$$

Given that backpropagation is the primary computational cost, our complexity analysis focuses on the count of backpropagation operations. In practice, $E_r$ and $E_d$ do not have to be large, and for the recovery phase, using only 1-2 sub-models can approximate well and yield satisfactory results.

**Increased Client Number & Large Client Scale:** According to the analysis of the computation cost, if the proportion of malicious clients remains unchanged while the total number of clients increases, the computational cost of our method would scale linearly. When the number of clients becomes large, the cost is indeed non-negligible. However, we would like to point out that all the operations

Table 10: Sensitivity of the distillation temperature.

| Temperature | 1 | | 2 | | 5 | | 10 | |
|---|---|---|---|---|---|---|---|---|
| | *ASR* | *GACC* | *ASR* | *GACC* | *ASR* | *GACC* | *ASR* | *GACC* |
| *FedSKU* | 3.35% | 75.40% | 3.20% | 73.52% | 2.91% | 76.68% | 3.06% | 74.92% |

are conducted on the server side. Considering that the servers usually have sufficient resources, it is acceptable to introduce some overhead on it. If computational resources are indeed limited, we can adopt the following strategy as a trade-off: (1) decreasing $E_r$, $E_d$, and $N_{sub}$; (2) monitoring the number of malicious clients and skipping some unlearning steps if the number is large.

### E.3 LIMITATIONS

First, our method may not extend to untargeted poisoning attacks, as they do not typically rely on a trigger pattern for targeted activation. Additionally, the computational demands of our unlearning process necessitate a server with sufficient resources. Finally, our method is based on the malicious-client detection techniques, and thus the accuracy of these detection methods may affect the performance of our approach. In the future, we will try our best to further address these limitations.

### E.4 SENSITIVITY OF THE DISTILLATION TEMPERATURE

We conducted experiments on CIFAR to assess the impact of various distillation temperature values. As illustrated in Table D, the distillation temperature exhibits minimal effects on both ASR and GACC, indicating the robustness of our method.

### E.5 DISCUSSION ON FEDRECOVER

The main differences between FedRecover Cao et al. (2023) and FedSKU are as follows:

Memory Demand: FedRecover requires substantial memory storage to maintain the historical information necessary for recovery. FedSKU, in contrast, does not impose such memory demands.

Communication Overhead: FedRecover necessitates additional communication rounds between clients and the server for update corrections, and FedSKU can operate without requiring extra communication overhead.

Performance Comparison: FedSKU can significantly improve accuracy for methods that directly exclude the backdoored models and conduct clean model aggregation, which exceeds the ACC upper bound of a model recovered by FedRecover, whose primary objective is being close to the train-from-scratch clean aggregated models.

In conclusion, while both FedSKU and FedRecover offer robust defense strategies, they are optimized under different objectives and constraints. FedSKU enhances the accuracy of the aggregated clean model, making it suitable for scenarios where accuracy is of high priority and server resources are abundant. FedRecover, on the other hand, can be applied for more general scenarios (i.e., not limited to backdoors) but will require high memory demand and communication cost for clients.

