# OpenReview forum: "FedSKU: Defending Backdoors in Federated Learning Through Selective Knowledge Unlearning"
_ICLR.cc/2024/Conference — Submitted to ICLR 2024_

### Official Review · Reviewer_d1FD · 2023-10-28

**Soundness:** 3 good
**Presentation:** 3 good
**Contribution:** 3 good
**Rating:** 8
**Confidence:** 3

**Summary:**

This paper proposes a new method called FedSKU (Federated Selective Knowledge Unlearning) to defend against backdoor attacks in federated learning. Compared to existing coarse-grained defenses that either completely remove suspected malicious models or add noise, FedSKU takes a more fine-grained approach by decomposing the model into the malicious trigger and useful knowledge. It recovers the trigger pattern using a novel pre-aggregation scheme for efficiency. Then it uses a dual distillation process to unlearn the trigger while preserving only clean knowledge in a surrogate model. This allows aggregating the useful knowledge from malicious models.

Experiments on image datasets like CIFAR-10/100 and Tiny ImageNet validate FedSKU. It improves accuracy by up to 6.1% over defenses like FLAME and Krum, with negligible increase in attack success rate (<0.01%). FedSKU also outperforms extensions of other unlearning methods like BAERASER and NAD to federated learning. Overall, FedSKU effectively utilizes knowledge from malicious models to improve accuracy while defending backdoor attacks.

**Strengths:**

1. Proposes a novel selective unlearning framework FedSKU that decomposes models into triggers and useful knowledge for fine-grained backdoor defense in federated learning.
2. Designs efficient techniques like pre-aggregation scheme for trigger recovery and dual distillation loss to selectively unlearn triggers while retaining useful knowledge.
3. Achieves significant accuracy gains over prior defenses like FLAME and Krum on CIFAR and Tiny ImageNet datasets, with marginal increase in attack success rate.
4. Outperforms extensions of other unlearning methods like BAERASER and NAD to federated learning scenario.
5. Comprehensive experiments analyzing impact of non-IID data, ratio of malicious clients etc.

**Weaknesses:**

1. Accuracy improvements are higher on CIFAR than Tiny ImageNet - I think more analysis are needed for why FedSKU works better on certain datasets and if this could be a sign of generalization difficulties.
2. No major limitations of the approach have been discussed.

**Questions:**

1. The pre-aggregation scheme for efficient trigger recovery makes sense intuitively, but more details or intuition could be provided on why aggregating backdoored models retains the malicious triggers reliably.
2. For unlearning using dual distillation, how sensitive is the performance to the hyperparameters like the distillation temperature? Was there any tuning done to set the hyperparameters?

---

> ### Author Response · Authors · 2023-11-17
> **Response to Reviewer d1FD**
>
> We sincerely thank you for your helpful and valuable feedback on our paper. We address your comments and questions below.
>
> **Weekness1:** Accuracy Analysis on Different Datasets.
>
> **Answer:** Thanks for your insightful question. We believe that generalization difficulties do not exist, as our method demonstrates performance improvement across all datasets, albeit to varying degrees. We speculate that the difference on the accuracy improvement can be attributed to the inherent complexity of the datasets. For example, the lower resolution and simpler feature space of CIFAR may facilitate more effective unlearning compared to the intricate Tiny-ImageNet, resulting in higher accuracy improvements.
>
> **Weekness2:** Limitations.
>
> **Answer:** First, our method may not extend to untargeted poisoning attacks, as they do not typically rely on a trigger pattern for targeted activation. Additionally, the computational demands of our unlearning process necessitate a server with sufficient resources. Finally, our method is based on the malicious-client detection techniques, and thus the accuracy of these detection methods may affect the performance of our approach. In the future, we will try our best to further address these limitations. The limitations have been added in the revision (details in Appendix E.3).
>
> **Question1:** Intuition on Why Aggregating Backdoored Models Retains the Malicious Triggers Reliably.
>
> **Answer:** The rationale behind pre-aggregation lies fundamentally in the attackers' goals. FL Attackers embed backdoor triggers in their updates with the intention that, once aggregated, the global model will inherit and manifest these backdoor characteristics. Pre-aggregation effectively simulates the aggregation process that would occur on the server, thus inherently preserving the malicious features introduced by the attackers.
>
> **Question2:** Sensitivity of the Distillation Temperature.
>
> **Answer:** We conducted experiments on CIFAR to assess the impact of various distillation temperature values. As illustrated in the following table, the distillation temperature exhibits minimal effects on both ASR and GACC, indicating the robustness of our method.
>
> | temperature | 1    |     | 2    |     | 5    |     | 10   |     |
> |-------------|------|-----|------|-----|------|-----|------|-----|
> |             | ASR  | GACC| ASR  | GACC| ASR  | GACC| ASR  | GACC|
> |   FedSKU    | 3.35% | 75.40%| 3.20%  |73.52%| 2.91% |76.68%| 3.06% |74.92%|
>
> We have added the experimental results in Appendix E.4.
>
> Thank you again for your valuable time spent reviewing our paper!

---

> > ### Author Response · Authors · 2023-11-23
> > **A kindly request for your response**
> >
> > We hope this message finds you well. We are writing to kindly request your response for our rebuttal. As the deadline for the discussion draws near (no more than 12 hours), We would greatly appreciate if you could let us know whether the rebuttal addresses your concerns. Thanks for your attention!

---

### Official Review · Reviewer_mcVc · 2023-10-31

**Soundness:** 3 good
**Presentation:** 2 fair
**Contribution:** 3 good
**Rating:** 5
**Confidence:** 4

**Summary:**

The paper proposes a backdoor defense technique in federated learning (FL) by
first identifying the possible trigger via trigger inversion, and then unlearn
the trigger from the model. The unlearning is done by distillation, assuming a
set of public dataset.

**Strengths:**

From writing aspect, the paper is easy to follow and understand. Its method description is clear.

Experiment show that the method is very effective, compared with existing
methods.

**Weaknesses:**

The used technique in this paper, trigger inversion and distillation, do to seem
to be significantly different from existing work. For example, its inversion
method is leveraging MESA (Qiao et al., 2019). I am not sure about the
novelty and significance of the technique.

There is no clear threat model in the paper. For example, both the trigger
recovery and unlearning require certain public data. But what types of public
data? There are various backdoors that work on a subset of inputs or outputs
labels. Does this method work on all these attacks? Based on my understanding, I
do not think it can cover all backdoor attacks. However, without a clear threat
model clarifying the assumptions, I have no information to leverage -- so does
the paper itself.

What does the method guarantee? Namely, will the proposed unlearning method be
"exact" or "approximate"?

Another line of work, e.g., FedRecover, that tries to recover from poisoning
attacks without the need to recover the trigger (and is also not limited to
backdoors), and also guarantees the recovered model is similar to the one
trained on non-poisoning data with a practical difference bound. The paper
should also include a discussion and comparison on that.

**Questions:**

See above.

---

> ### Author Response · Authors · 2023-11-17
> **Response to Reviewer mcVc (1/2)**
>
> Thank you for your careful review and valuable feedback. We hope the following clarification can address your concerns.
>
> **Weekness1:** Technical Significance.
>
> **Answer:** First, we are the first to introduce the idea of utilizing the benign knowledge embedded in the backdoored models to enhance the FL performance. Second, for the proposed techniques, we would like to highlight that we have made substantial modifications to the existing techniques, rather than directly applying them. Specifically, for trigger recovery, we introduce a pre-aggregation scheme to the backdoored models before conducting the trigger recovery, in order to avoid the massive computational consumption. For the distillation techniques, Section 4.3 has explicitly described our key designs (i.e., distillation architecture and distillation loss) to achieve unlearning under the FL scenario, which is fundamentally different from the traditional machine unlearning techniques. We will highlight these distinctions more explicitly in our revision to clarify the unique contributions of our work.
>
> **Weekness2:** Threat Model.
>
> **Answer:** We are truly grateful for your reminder regarding the threat model. Here we formally define it.
>
> *Threat Model:* In our paper, we follow the attack setting of the previous FL defense work [1]. The attacker $\mathcal{A}$ intends to inject the backdoor pattern into the aggregated model, which will subsequently affect all $N$ participants. The attacker controls $m < \frac{N}{2}$ clients and utilizes them to upload the elaborate model updates to the server. The attacker has full knowledge of the clients under his control, including the local training dataset, training processes, and model parameters. However, the attacker has no information about all other clients and no control of the server. The attacker intends to poison the global model $G$ and simultaneously makes it behave as if it is a normal model. Specifically, the attacker's goal is:
>
> \begin{equation}
> \forall x' \in T, G(x') = y_{\text{Att}}
> \end{equation}
> and
> \begin{equation}
> \forall x_i \in D, G(x_i) = y_i.
> \end{equation}
>
> Here, $y_{\text{Att}}$ is the attacker's target label, $y_{i}$ denotes the correct label of input $x_{i}$, $D$ is the clean data and $T$ denotes the data containing the backdoor pattern chosen by the attacker.
>
> In addition, regarding the public data used in the trigger recovery and unlearning process, our method requires a small subset of clean data with the same data type as our training data. Consistent with the settings used in NAD [2], we only utilize 5% data of the dataset as the public data.
>
> As for the backdoor types, our method is effective against typical backdoor attacks that target a subset of inputs, meaning that the backdoor is activated only when the input data contains a specific trigger pattern.
>
> We have also added the threat model in the revision (details in Appendix E.1).
>
> [1] Nguyen, Thien Duc, et al. "{FLAME}: Taming backdoors in federated learning." 31st USENIX Security Symposium (USENIX Security 22). 2022.
>
> [2] Li, Yige, et al. "Neural attention distillation: Erasing backdoor triggers from deep neural networks." ICLR 2021. 2021.
>
> **Weekness3:** Method Guarantee.
>
> **Answer:** Due to the poor interpretability of DNN models, our approach cannot achieve an "exact unlearning," which means that we can accurately locate all the backdoors. Instead, we assure an "approximate unlearning" by our proposed dual distillation method.

---

> ### Author Response · Authors · 2023-11-17
> **Response to Reviewer mcVc (2/2)**
>
> **Weekness4:** Discussion on FedRecover [3].
>
> **Answer:** The main differences between FedRecover and FedSKU are as follows:
>
> *Memory Demand:* FedRecover requires substantial memory storage to maintain the historical information necessary for recovery. FedSKU, in contrast, does not impose such memory demands.
>
> *Communication Overhead:* FedRecover necessitates additional communication rounds between clients and the server for update corrections, and FedSKU can operate without requiring extra communication overhead.
>
> *Performance Comparison:* FedSKU can significantly improve accuracy for methods that directly exclude the backdoored models and conduct clean model aggregation, which exceeds the ACC upper bound of a model recovered by FedRecover, whose primary objective is being close to the train-from-scratch clean aggregated models.
>
> In conclusion, while both FedSKU and FedRecover offer robust defense strategies, they are optimized under different objectives and constraints. FedSKU enhances the accuracy of the aggregated clean model, making it suitable for scenarios where accuracy is of high priority and server resources are abundant. FedRecover, on the other hand, can be applied for more general scenarios (i.e., not limited to backdoors) but will require high memory demand and communication cost for clients.
>
> We have added the discussion in Appendix E.5.
>
> [3] Cao, Xiaoyu, et al. "Fedrecover: Recovering from poisoning attacks in federated learning using historical information." 2023 IEEE Symposium on Security and Privacy (SP). IEEE, 2023.
>
>
> Thank you again for your valuable time spent reviewing our paper!

---

> > ### Author Response · Authors · 2023-11-23
> > **A kindly request for your response**
> >
> > We hope this message finds you well. We are writing to kindly request your response for our rebuttal. As the deadline for the discussion draws near (no more than 12 hours), We would greatly appreciate if you could let us know whether the rebuttal addresses your concerns. Thanks for your attention!

---

### Official Review · Reviewer_FcGK · 2023-11-01

**Soundness:** 3 good
**Presentation:** 3 good
**Contribution:** 2 fair
**Rating:** 5
**Confidence:** 4

**Summary:**

This paper addresses the vulnerability to backdoor attacks that manipulate model parameters to deceive the aggregation process. Unlike existing defenses that employ coarse-grained methods, this research takes a more nuanced approach. The authors propose a novel technique called FedSKU, which involves decomposing the uploaded model into two distinct components: malicious triggers and useful knowledge.

**Strengths:**

- The concept of selective unlearning represents a novel and compelling advancement in comparison to the coarser-grained defenses commonly used.
- Through extensive evaluation across various datasets, the method demonstrates superior performance in accuracy compared to state-of-the-art defenses, all while keeping the increase in attack success rate at a negligible level.
- The inclusion of convergence analysis and ablation studies offers valuable insights into the inner workings of the method.

**Weaknesses:**

- The method proposed hinges on anomaly detection for identifying malicious clients. It's important to note that this approach has its limitations; attackers may find ways to evade detection. The authors should delve deeper into this aspect for a more comprehensive discussion.
- The paper lacks an in-depth analysis of the computational overhead associated with trigger recovery and unlearning.
- The experiments conducted on non-iid settings are not as extensive as one might expect.

**Questions:**

- The effectiveness of the unlearning process can be influenced by various factors, including model complexity, available data volume, and the complexity of the information to be forgotten. How can we ensure that unlearning remains effective after anomaly detection?
- How does the computational overhead of the trigger recovery and unlearning process change as the number of clients increases?
- Instead of performing unlearning, wouldn't it be more efficient to simply exclude the detected malicious clients from the training process?
- Given the potentially large number of participating clients in Federated Learning, how can we guarantee that the proposed method remains effective despite the high computation overhead?

---

> ### Author Response · Authors · 2023-11-17
> **Response to Reviewer FcGK (1/2)**
>
> Thank you for your careful review and valuable feedback. We hope the following clarification can address your concerns.
>
> **Weakness1:** Discussion on anomaly detection.
>
> **Answer:** We agree with the reviewer that advanced adversaries may manage to bypass detection mechanisms. However, what we want to emphasize is that anomaly detection, as a defense mechanism in Federated Learning, is a topic where researchers continually explore more advanced and effective mechanisms to resist relevant attackers. Many recent detection-based defense mechanisms, such as FLAME[1], DeepSight[2], and FLDetector[3], have demonstrated increased maturity, achieving near-perfect accuracy under specific attack scenarios. Therefore, we believe that with the continuous improvement of anomaly detection mechanisms, our finer-grained enhancements based on it will also remain meaningful.
>
> [1] Nguyen, Thien Duc, et al. "{FLAME}: Taming backdoors in federated learning." 31st USENIX Security Symposium (USENIX Security 22). 2022.
>
> [2] Rieger, Phillip, et al. "Deepsight: Mitigating backdoor attacks in federated learning through deep model inspection." NDSS 22. 2022.
>
> [3] Zhang, Zaixi, et al. "FLDetector: Defending federated learning against model poisoning attacks via detecting malicious clients." Proceedings of the 28th ACM SIGKDD Conference on Knowledge Discovery and Data Mining. 2022.
>
> **Weakness2& Question2& Question4:** Analysis of the computational overhead for our approach.
>
> **Answer:** Our approach introduces an unlearning process compared to conventional removal-based defense, which comprises a trigger recovery part and an unlearning part. In the following parts, we analyze the overhead in detail.
>
> *Trigger Recovery:* In the trigger recovery phase, we start by searching for the attacker's target label. We denote the label currently being searched for as $i$ and the number of the label is *num\_class* (denoted as $N_{c}$). We then apply our recovery process to obtain multiple sub-models, denoted by $G_{i}$, where $i = 1, \ldots, N_{sub}$. For each sub-model, we train for a number of epochs defined by *num\_recovery\_epoch* (abbreviated as $E_{r}$) according to Equation (1). Assuming the dataset size at the server is $V$ and the batch size during training is *recovery\_batch\_size* (abbreviated as $BS_{r}$), the complexity of the recovery part can be described as performing $N_{c} \times E_{r} \times N_{sub} \times \frac{V}{BS_{r}}$ backpropagations.
>
> *Unlearning:* After obtaining the final set of sub-models, $G = [G_{1}, G_{2}, \ldots, G_{N_{sub}}]$, we proceed to conduct unlearning (i.e., distillation). This involves conducting *distill\_epoch* (abbreviated as $E_{d}$) training epochs with Equation (4) for unlearning. The batch size for distillation is denoted as *distill\_batch\_size* (abbreviated as $BS_{d}$), leading to a distillation complexity of $E_{d} \times \frac{V}{BS_{d}}$ backpropagations.
>
> *Total overhead:* Let us denote the number of detected anomalous clients as *abnormal\_client\_num* (abbreviated as $N_{a}$). Taking into account the pre-aggregation strategy, the overall complexity of our approach for the detected anomalous clients can be expressed as follows:
>
> \begin{equation}
> \text{Total Overhead } = N_{c} \times E_{r} \times N_{sub} \times \frac{V}{BS_{r}} + N_{a} \times (E_{d} \times \frac{V}{BS_{d}})
> \end{equation}
>
> Given that backpropagation is the primary computational cost, our complexity analysis focuses on the count of backpropagation operations. In practice, $E_{r}$ and $E_{d}$ do not have to be large, and for the recovery phase, using only 1-2 sub-models can approximate well and yield satisfactory results.
>
> *Increased Client Number (Question2) & Large Client Scale (Question4):*
> According to the analysis of the computation cost, if the proportion of malicious clients remains unchanged while the total number of clients increases, the computational cost of our method would scale linearly. When the number of clients becomes large, the cost is indeed non-negligible. However, we would like to point out that all the operations are conducted on the server side. Considering that the servers usually have sufficient resources, it is acceptable to introduce some overhead on it. If computational resources are indeed limited, we can adopt the following strategy as a trade-off: (1) decreasing $E_{r}$, $E_{d}$, and $N_{sub}$; (2) monitoring the number of malicious clients and skipping some unlearning steps if the number is large.
>
> We have also added the related description in the revision (details in Appendix E.2).

---

> ### Author Response · Authors · 2023-11-17
> **Response to Reviewer FcGK (2/2)**
>
> **Weakness3:** Experiments on extensive non-iid settings.
>
> **Answer:** To explore the more extensive non-iid settings, we added an experiment on CIFAR-10 for different methods on the non-iid degree of 0.8, which is considered as a high non-iid situation. As shown in the following table （we also updated Table 4 in our revision）, when the degree reaches 0.8, both ASR and GACC are largely affected for all methods, which means that existing methods cannot cope with an extreme non-iid situation. We would like to highlight that all the defense methods are not specifically designed for the non-iid scenario and if we want to conduct evaluation on this condition, some strategies of addressing the non-iid problem should be combined for each method to make a fair comparison, which is out of our scope.
>
> | Non-iid | 0.2    |      | 0.4  |      | 0.6  |      | 0.8  |      |
> |---------|--------|------|------|------|------|------|------|------|
> |         | ASR    | GACC | ASR  | GACC | ASR  | GACC | ASR  | GACC |
> | BAERASER| 6.97%   | 69.01%| 14.22%| 65.27%| 9.79% | 51.14%| 13.16%| 33.88%|
> | NAD     | 15.75%  | 68.80% | 18.41%| 64.55%| 30.97%| 52.98%| 16.24%| 32.53%|
> | FLAME   | 2.66%   | 67.54%| 3.60%  | 58.94%| 4.74% | 44.93%| 11.82%| 22.83%|
> | Ours    | 2.96%   | 68.12%| 2.84% | 62.44%| 4.20%  | 49.45%| 12.04%| 25.44%|
>
> **Question1:** Effectiveness of the Unlearning Process.
>
> **Answer:** Thanks for your question, We would like to address each of the factors you've highlighted.
>
> *Model Complexity:* We acknowledge that higher model complexity could pose challenges for the unlearning process, potentially requiring more epochs and increased computational overhead. However, it is worth noting that in FL, models are often less complex due to the inherent difficulties of training over distributed resource-constrained clients. Moreover, for attackers, embedding backdoors into highly complex models is equally challenging, which naturally increases the difficulty of the attack.
>
> *Available Data Volume:* In our unlearning process, we follow the NAD method to utilize 5% of the training data, which has proven to be sufficient for maintaining the effectiveness of our approach. We believe that this data volume is acceptable and realistic for most real-world scenarios.
>
> *Complexity of Information to be Forgotten:* We agree that the complexity of information could affect the unlearning process. However, this complexity primarily impacts specific unlearning algorithms, i.e., how to achieve unlearning in the presence of more intricate data information. This paper proposes a design strategy (dual distillation) for unlearning and leverages distillation algorithms for implementation. In the context of more complex data scenarios, we can use more advanced distillation algorithms to align with our approach, such as employing sophisticated distillation loss.
>
> **Question3:** Simple Exclusion of Malicious Clients.
>
> **Answer:** While exclusion might seem straightforward, it becomes ineffective when the proportion of malicious clients is significant. Direct exclusion in such cases would lead to the great loss of substantial useful knowledge embedded in the backdoored models, negatively impacting the global model's accuracy. The baseline FLAME can be considered as a simple exclusion method.
>
> Thank you again for your valuable time spent reviewing our paper!

---

> > ### Comment · Reviewer_FcGK · 2023-11-22
> >
> > I appreciate the authors' efforts in addressing my concerns.
> >
> > I remain unconvinced by the argument favoring unlearning instead of straightforwardly excluding malicious clients. In response to Reviewer mcVc, this study focuses on "approximate unlearning." What if the same malicious clients persistently launch attacks in subsequent rounds, even after being subject to approximate unlearning in the previous round? The impact of the attack's footprint may endure, affecting the training process if an exact unlearning cannot be achieved.
> >
> > Continuing the training process when a substantial number of malicious clients are present might lead to significant consumption of computational resources without generating meaningful improvements in the model. It could be more resource-efficient to allocate these resources to addressing security concerns and rectifying the compromised learning environment.
> >
> > Removing malicious clients stands out as a straightforward and robust solution. This approach adheres to the principle of simplicity and diminishes the complexity of the FL system.

---

> > > ### Author Response · Authors · 2023-11-22
> > > **Response to the feedback**
> > >
> > > Thank you for your response. We would like to further clarify the raised issues.
> > >
> > > First, as for the concern that "approximate unlearning" may result in enduring the attack's footprint, we would like to highlight that our experiment setup exactly involves the attackers in every round and the experiment results of low ASR and improved GACC indicate that even in this setting, **our approximate unlearning can effectively eliminate the attack's footprint continuously.** Besides, the premise for the accumulation of persistent attacks is that each round of attacks must yield noticeable effects. However, in our scenario, **each unlearning round almost entirely eliminates the impact of the attack, resulting in the generation of a global model with a very low ASR.** Therefore, persistent attacks are ineffective under our proposed technique. We hope that these experimental results and mechanism analyses can address your concerns.
> > >
> > > Second, continuing the training process in the presence of a significant number of malicious clients can result in the slow convergence of remove-based methods. In contrast, although our method introduces some computational cost, **it does generate meaningful improvements**. As demonstrated in the convergence curve for CIFAR-100 in Figure 3, our method achieves comparable GACC after 120 epochs as the removal-based method did after 320 epochs, achieving nearly 3X faster convergence.
> > >
> > > Finally, **the core principle of FL is to aggregate more knowledge under the umbrella of privacy preservation.** Simple exclusion is straightforward but has limitations in losing significant effective knowledge. Our approach provides a well-balanced trade-off, substantially increasing the aggregated knowledge with negligible ASR, which is highly meaningful for accomplishing the goal of FL.
> > >
> > > Once again, we appreciate your feedback and hope that our clarifications can address your concerns. Please let us know if you have any further questions or comments.

---

> > > > ### Author Response · Authors · 2023-11-23
> > > > **A kindly request for your response**
> > > >
> > > > We hope this message finds you well. We are writing to kindly request your response for our rebuttal. As the deadline for the discussion draws near (no more than 12 hours), We would greatly appreciate if you could let us know whether the rebuttal addresses your concerns. Thanks for your attention!

---

### Official Review · Reviewer_wRHA · 2023-11-03

**Soundness:** 3 good
**Presentation:** 3 good
**Contribution:** 3 good
**Rating:** 6
**Confidence:** 2

**Summary:**

The authors propose FedSKU, a defense mechanism that detects and selectively unlearns harmful backdoors in uploaded models. They introduce a pre-aggregated trigger recovery scheme to efficiently train a trigger pattern generator, reducing training overhead in the FL system. They also designed a dual distillation method for selective knowledge unlearning.

**Strengths:**

1. The paper emphasizes that pre-aggregation in the model inevitably preserves certain malicious features. They designed the method based on this insight that can effectively reduce the Federated Learning (FL) training overhead from the trigger generator.
2. The suggested method extracts valuable knowledge from backdoored clients, providing a novel defense method against backdoors while preserving competitive global accuracy.
3. The experiments are comprehensive, encompassing varying numbers of malicious clients, initialization parameters, and convergence comparisons among others.

**Weaknesses:**

1. The "pre-aggregated" process lacks clarity. Also, it would be better to discuss the federated aggregation method on the main page rather than in the appendix.

2. While it's acceptable that FEDSKU has a marginally lower GACC compared to DNN unlearning methods like BAERASER and NAD, given their excessively high ASR indicates unsuccessful defense, it would strengthen the claim about ‘'take the essence and discard the dross' if the author could elucidate why DNN unlearning methods have superior GACC.

**Questions:**

1. In Figure 2, it appears that two backdoored models exist, but only one is detected. This setup could be confusing due to the absence of the "pre-aggregates" step in the Trigger recovery process. It might be clearer if the framework depicted the detection of multiple backdoored models, thereby illustrating the details of "pre-aggregates" and showing that each backdoored model has a corresponding surrogate.

---

> ### Author Response · Authors · 2023-11-17
> **Response to Reviewer wRHA**
>
> Thank you very much for the positive feedback and valuable comments. We hope the following clarification can address your concerns.
>
> **Weakness1& Question1:** Issues on pre-aggregation and federated methods.
>
> **Answer:** We apologize for the lack of clarity. The "pre-aggregated" process mainly includes the following steps: (1) detecting the backdoored models with existing detection methods; (2) employing the aggregation algorithm FedAvg to average the parameters among these backdoored models, such that we can obtain a global model. Note that the global model is merely used as the target for trigger recovery rather than the practical model aggregation. In this way, we can significantly reduce the overhead since only a single model is required to conduct the recovery. At the same time, due to the inherent robustness of FL attacks to the aggregation process, we believe that pre-aggregation will not eliminate the backdoor.
>
> In addition, we have modified Figure 2 to accurately depict the scenario where multiple backdoored models exist and clearly illustrate the "pre-aggregation" step in the trigger recovery process. We sincerely thank you for your observation regarding Figure 2. Furthermore, we have transferred the discussion of the federated aggregation methods in the appendix to the main page. Please refer to our revision.
>
> **Weakness2:** Reasons about why DNN unlearning methods have superior GACC.
>
> **Answer:** Thanks for your valuable suggestion. Generally, there is a significant possibility that unlearning may unlearn some useful features if we adopt an aggressive unlearning strategy, which will result in a reduction in accuracy. In our approach, we employ dual distillation to achieve unlearning. While it effectively eliminates the backdoor, the extent of unlearning is larger than simple distillation methods such as NAD and BAERASER. Consequently, our GACC exhibits a slight reduction compared to baseline methods. However, this dual distillation achieves a significantly lower ASR than NAD and BAERASER, which indicates a more effective defense method.
>
> Thank you again for your valuable time spent reviewing our paper!

---

> > ### Author Response · Authors · 2023-11-23
> > **A kindly request for your response**
> >
> > We hope this message finds you well. We are writing to kindly request your response for our rebuttal. As the deadline for the discussion draws near (no more than 12 hours), We would greatly appreciate if you could let us know whether the rebuttal addresses your concerns. Thanks for your attention!

---

### Author Response · Authors · 2023-11-17
**Global Response**

Dear AC and reviewers,

We sincerely appreciate the time and effort you have dedicated to reviewing our submission. In response to your insightful comments, we have diligently revised and enhanced our manuscript, incorporating the following additional discussions and experiments:

1.	We have included the pre-aggregation process in Figure 2 and introduced an additional non-iid setting in Table 4.

2.	Appendix E now contains details about the threat model, experimental results on distillation temperature, and a discussion on limitations and related works.

3.	We have highlighted the technical significance and novelty of our work.

4.	The settings for our method, originally located in the appendix, have been transferred to the main text.

For your convenience, these updates are temporarily highlighted in "red" for your review. We hope that our response and revisions can effectively address all concerns raised by the reviewers.

Thank you very much for your time and consideration.

Best regards,

Paper4512 Authors

---

### Meta-Review · Area_Chair_6LdP · 2023-12-09

**Metareview:**

The paper is appreciated for its innovative approach to addressing backdoor attacks in federated learning systems. FedSKU's methodology of selective unlearning is seen as a significant advancement over existing coarser-grained defenses. The experiments are comprehensive and demonstrate the method's effectiveness in various settings.

However, there are notable concerns. The clarity of certain processes, such as the "pre-aggregated" step, needs improvement. Reviewers also express a need for a more in-depth analysis of the computational overhead, especially considering the scalability with an increasing number of clients. Additionally, the variation in performance across different datasets and the absence of a detailed discussion on major limitations of the approach are pointed out as areas needing further exploration.

Despite its novel approach and comprehensive testing, FedSKU's current iteration has notable gaps in clarity, performance comparison, computational analysis, and generalization. These issues are critical enough to warrant a rejection at this stage. However, addressing these concerns in future work could significantly improve the paper's contribution to the field of federated learning and backdoor defense. The innovative core of the paper is recognized, but refinement and further development are necessary to meet the high standards required for acceptance.

**Justification For Why Not Higher Score:**

While the paper introduces an innovative approach to addressing backdoor attacks in federated learning, significant concerns raised by the reviewers limit its current acceptability.

Lack of Clarity in Key Processes: Reviewer wRHA highlighted that the "pre-aggregated" process, a critical component of the FedSKU methodology, lacks clarity. This obscurity in the method's execution undermines its comprehensibility and applicability.

Comparative Performance Issues: Concerns were raised about FedSKU's marginally lower global accuracy compared to existing DNN unlearning methods like BA ERASER and NAD. This comparative underperformance in crucial metrics suggests that the method may not offer a substantial improvement over current state-of-the-art approaches.

Computational Overhead and Scalability: Reviewer FcGK pointed out the absence of a thorough analysis of the computational overhead associated with FedSKU, particularly important in the context of federated learning with potentially numerous clients. This omission raises doubts about the method's practical scalability and efficiency.

Limited Experimentation in Non-IID Settings: The paper's experiments in non-IID settings were noted to be less extensive than expected, which is a significant consideration in federated learning environments.

Generalization Concerns: Reviewer d1FD observed that accuracy improvements with FedSKU were higher on certain datasets like CIFAR than on others like Tiny ImageNet. This inconsistency suggests potential issues with the method's generalizability across diverse datasets.

**Justification For Why Not Lower Score:**

NA

---

### Decision · Program_Chairs · 2024-01-16

Reject